# Determinants of stunting among adolescent girls in schools of Digo Tsion Town, Northwest Ethiopia: Unmatched case control study

Ambaw Abebaw Emrie[1]*, Getasew Tesfa[2], Yeneneh Ayalew[2], Adugnaw Bantie Kebie[3], Tamene Fetene Terefe[1], Agerie Aynalem Mewahegn[1], Bogale Chekole[1], Fisha Alebel GebreEyesus[1], Legese Fekede Abza[1], Selamsew Kindie Nega[4]

1 Department of Nursing, Wolkite University, Wolkite, Ethiopia, 2 Department of Pediatrics and Child Health Nursing, Bahir Dar University, Bahir Dar, Ethiopia, 3 Department of Pediatrics and Child Health Nursing, Debre Markos University, Debre Markos, Ethiopia, 4 Department of Nursing, Mada Walabu University, Bale Robe, Ethiopia

* ambabebaw@gmail.com

## Abstract

### Introduction

Stunting is a height-for-age (Z-score) less than minus two standard deviations below the mean of reference standard. It is the most important sign of long-term chronic undernutrition and public health problem in Ethiopia. However, little information was known regarding determinants of stunting among adolescents since it had mostly been investigated in late infancy, especially among children under the age of five. Therefore, identifying determinants of stunting among adolescent girl is still crucial.

### Objective

To identify determinants of stunting among adolescent girls in schools of Digo Tsion Town, Northwest Ethiopia, 2022.

### Methods and materials

Case-control study was conducted among 417 adolescent girls (104 cases and 313 controls) in schools of Digo Tsion Town with computer generated simple random sampling technique. World Health Organization Anthroplus 2007 software was used for analyzing anthropometrics data. Data was collected by epicollect5 mobile application through interview by using structured questionnaire. The data was entered in epi data 4.6 and exported into Statistical Package for Social Science version 26. Variables with p- value $\leq$ 0.25 in bivariable analysis were candidate for multivariable analysis. Model fitness was checked by Hosmer and Lemon Show fitness of test. Variables having a P-value < 0.05 in multivariable analysis were declared as statistically significant at 95% CI. The result was presented by statement, figures, and tables.

**Data Availability Statement:** All relevant data are within the manuscript and its Supporting Information files.

**Funding:** The author(s) received no specific funding for this work.

**Competing interests:** The authors have declared that no competing interests exist.

**Abbreviations:** AOR, Adjusted odds ratio; BMI, body mass index; COR, crude odds ratio; EDHS, Ethiopian demographic and health survey; GDHS, Ghana demographic and health survey; HEW, health extension worker; NCD, non communicable disease; NNP, national nutritional program; SPSS, statistical package for social science; WHO, World Health Organization.

## Results

A total of 409 (100 cases and 309 controls) adolescent girls participated, with a response rate of 96% for cases and 98.72% for controls. Food insecurity (AOR = 2.13, CI [1.15, 3.93]), low dietary diversity score (AOR = 1.99, CI [1.06, 3.73]), drinking coffee/tea immediately while eating meals (AOR = 2.19, CI [1.22, 3.95]), not getting nutritional counsel (AOR = 2.07, CI [1.17, 3.66]), chronic illness (AOR = 3.78, CI [1.16, 12.3]), and not visited by health extension workers at home (AOR = 1.85, CI [1.03, 3.31]) were statistically significant determinants of stunting.

## Conclusion

Stunting among adolescents is influenced by a low dietary diversity score, a food-insecure household, drinking coffee or tea immediately while eating a meal, not receiving nutritional counseling, having a chronic illness, and not being visited by health extension workers at home. Future researchers would do better to undertake prospective studies. Health extension workers are better able to provide nutritional counsel for adolescent.

## Introduction

Stunting is defined as having a height-for-age z-score (HAZ) that is less than minus two standard deviations (-2 SD) below the mean of a reference standard, and those who are less than minus three standard deviations (-3 SD) are termed severely stunted [1]. It is the most important sign of long-term chronic undernutrition, indicating a failure to achieve linear development as a result of protracted food deprivation and diseases during childhood [2].

Adolescents are defined by the United Nations and the World Health Organization as those between the ages of 10 and 19. Globally, there are an estimated 600 million adolescent girls [3]. For adolescent girls, a range of body and social changes take place during and following puberty, and it is a time of intense physical, psychosocial, and cognitive development [4].

Children and adolescents, in particular, are more susceptible to sickness and mortality as a result of malnutrition [5]. The pace of progress in addressing all forms of malnutrition is inexcusably sluggish. Although there has been significant progress in reducing early childhood stunting, the number of stunted individuals remains at 150.8 million all over the World [6].

Nutrition during adolescence plays an important role in the individual's life because they gain up to 50% of their adulthood weight, more than 20% of their adulthood height, and 50% of their adulthood skeletal mass [7]. And also, a sustained healthy diet during the adolescent period has the potential to limit harmful behaviors contributing to the epidemic of non-communicable diseases (NCDs) in adulthood [8].

Adolescent girls are vulnerable to stunting due to internal and external influences such as peer pressure, the desire to fit in among friends, and aggressive food marketing [9]. Almost one-third of the world's population is currently malnourished, and this is one of the most serious issues confronting the global community's development [10]. According to a study conducted in Pakistan 22.7%, West Java, Indonesia 48%, and West Bengal, India 54% of adolescent girls suffer from stunting [11–13].

The effect of stunting includes poor health and school performance, impaired physical and mental development, and perpetuation of the cycle of poverty, as it may result in deficits in productivity in adulthood [14]. Reducing stunting is an important part of the global health

promotion program policy. Despite likely changes in risk factors as children age, determinants of stunting are typically analyzed without taking into account age-related heterogeneity [15].

Based on the Ethiopia Nutrition Profile, approximately 22% of women of reproductive age are malnourished, which puts their children at risk for low birth weight, short stature, lower infection resistance, and a higher risk of sickness and mortality [16]. Children and adolescents who are stunted are unable to reach their full physical and mental potential [17]. Malnutrition, frequent illnesses, and a lack of social stimulation are the most common causes of stunting [18, 19]. Short-statured women are a significant risk factor for giving birth of stunted babies [20].

Adolescent health and nutrition interventions are crucial to achieving the 2030 Sustainable Development Goals, especially for adolescent girls [21]. One of the Ethiopian government's national nutritional program II strategic goals is to reduce adolescent malnutrition in all of its forms [22].

However, little information was known regarding stunting in adolescents because it had mostly been investigated in late infancy, especially among children under the age of five. In order to prevent chronic malnutrition, notably stunting in the population, the second window of opportunity after the first 1000 days was to analyze adolescent nutritional status and the determining factors [23]. And also, assessing the determinants of stunting among adolescent girls is important to address the problems of the coming generation [24]. Therefore, this study was aimed at identifying determinants of stunting among adolescent girls in the schools of Digo Tsion Town, Northwest Ethiopia.

## Methods and materials

### Study design and setting

A school-based case control study was conducted from May 13—June 13, 2022. The study was conducted in the schools of Digo Tsion Town (administrative Town of Bibugn Wereda), which is located 365 km from Addis Ababa to the northwest and 142 km from Bahir Dar City. The Town has two primary schools, one high school, and one preparatory school. There was a total of 7520 students, of whom 5232 were adolescent students and 2556 were adolescent girls, in all schools of the town registered for the academic year 2022, "Bibugn Wereda Education Office Report [unpublished]".

### Source and study population

Cases: All adolescent girls in the schools of Digo Tsion Town with a height for age of less than -2 Standard Deviation (Z score).

Controls: All adolescent girls in the schools of Digo Tsion Town with a height for age ≥ -2 Standard Deviation (Z score).

### Study unit

Adolescent girl

### Inclusion criteria

Cases: School adolescent girls with height for age less than -2SD and attended during data collection were included.

Controls: School adolescent girls with height for age ≥ -2SD and attended during data collection were included.

## Exclusion criteria

School adolescent girls with spinal curvature, third trimester pregnant, and adolescents walked with a wheelchair were excluded for both cases and controls.

## Sample size determination

To determine the sample size, significant predictors from different literatures were considered. Accordingly, the sample size was determined by considering a double population proportion formula by using Epi-info version 7 statistical package and assuming a two-sided significant level (α) of 5%, a 95% confidence level, power of 80%, and a 1:3 ratio of cases to controls. Drinking water source, residence, and food security status were used as significant predictors of stunting in the most recent studies. Sample size from food security was selected since it gives the optimal sample size (Table 1).

After adding 10% non-response rate = 38, the largest sample size was ($N_f$ = 417). Finally, 104 cases and 313 controls were selected as the final sample size for the study.

## Sampling technique and procedures

A computer- generated simple random sampling method was employed to select study participants. The height of each adolescent girl was measured by a stadiometer and recorded with their ages. And then, cases and controls were identified based on the World Health Organization Anthroplus 2007 software. A code was given for each adolescent girl, and a sampling frame was prepared in computer with their school ID, section code, and Z–score values. Then, one hundred four (104) cases and 313 controls that fulfilled inclusion criteria were selected. Proportional allocation was done to select the number of female adolescent students from each school (Fig 1).

## Dependent variable

Stunting (yes/no)

## Independent variables

Sociodemographic factors: age, marital status, residence, family size, wealth index, mother's education and occupation, father's education and occupation.

Dietary and nutritional service plus information factors: meal per day, decision-maker, coffee/tea drinking habit, skipping regular meals, snack use, usual food source, dietary diversity score, food security, nutritional counsel and using mass media.

Health and health service-related factors: receiving deworm tablet, home-to-home visit of HEW, menstrual status, malaria, chronic illness and anemia.

**Table 1. Sample size calculation for determinants of stunting among adolescent girls in schools of Digo Tsion Town, Northwest Ethiopia, 2022.**

| Variables | | $P_1$, $P_2$ and OR | Sample size |
|---|---|---|---|
| Residence [25] | Rural | $P_1$ = 68%, $P_2$ = 47.21% and | N = 235 |
| | Urban | OR = 2.38 | |
| Food security [26] | Insecure | $P_1$ = 59.2%, $P_2$ = 42.6% and | N = 379 |
| | Secure | OR = 1.951 | |
| Drinking water source [27] | Pipe & protected | $P_1$ = 43.1%, $P_2$ = 21.31% and OR | N = 182 |
| | River | = 2.80 | |

$P_1$: the percent of case exposed, $P_2$: the percent of control exposed, **OR:** Odds ratio

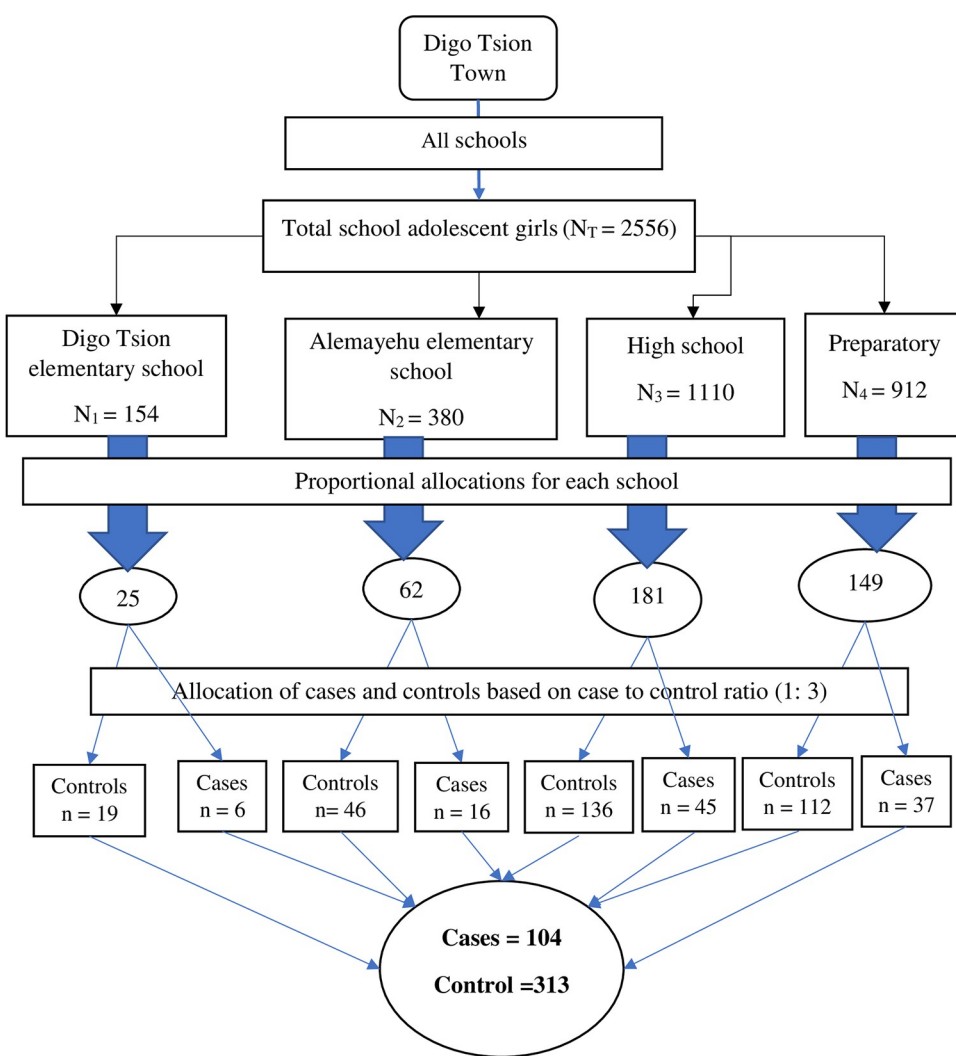

$N_T$: Total number adolescent girls

$N_1$: number of adolescent girls in Digo Tsion elementary school

$N_2$: number of adolescent girls in Alemayehu elementary school

$N_3$: number of adolescent girls in high school

$N_4$: number of adolescent girls in preparatory school

n: number of sample adolescent girls from each school

**Fig 1. Schematic presentation of sampling procedure for the determinants of stunting among adolescent girls in schools of Digo Tsion Town, Northwest Ethiopia, 2022.**

Personal hygiene, sanitation and environmental factors: home latrine, waste disposal, drinking water source, water purification methods, hand washing, using soap/detergent for hand washing.

## Operational definitions

**Stunting:** Is a chronic malnutrition which is height for age less than -2SD (Z score) from the WHO reference population [4].

**Decision-maker:** An individual who makes a decision for nutritional service in family member [28].

**Food secure household:** Were considered the availability of foods in homes in which every family member can access to consume, did not live-in hunger or fear of starvation in the households and answered the question for the presence of food worrying as 'no or rarely'. A food secure household had score of < 2 out of twenty seven for the food security assessment scale [29, 30].

**Food insecure household:** Were considered when the households sometimes or often worried about not having enough food and replied "yes" to any one of the remaining questions in the food security assessment tools. A food insecure household had a score ≥ 2 out of twenty seven for the food security assessment scale [29, 30].

**Low dietary diversity score:** Eating ≤ 4 groups of food from ten groups of foods in the last 24 hours [22, 31].

**Adequate dietary diversity:** Eating >4 groups of food from ten groups of foods in the last 24 hours [22, 31].

**Improved water source:** Water from piped water in home, public taps, protected dug wells, protected springs and rainwater [32].

**Unimproved water source:** Water from unprotected dug wells, unprotected springs and surface water/river water [32].

**Wealth index:** is a composite measure of a household's cumulative living standard. The wealth index is calculated using easy-to-collect data on a household's ownership of selected assets, such as televisions and bicycles; materials used for housing construction; and types of water access and sanitation facilities [32].

## Data collection tool and procedure

After reviewing different studies and reading supporting guidelines, a structured questionnaire was adapted. The questionnaire was prepared in English version and translated to Amharic, which was used for communication in the local community, and then retranslated back to English by language experts. The questionnaire was a collection of nine questions on socio-demographic characteristics, eight (8) questions to collect information about personal hygiene, sanitation, environmental factors, health-related information, nutritional services, and dietary-related factors. The questionnaire also comprises 10 questions that can assess the dietary diversity score, 18 questions for food security assessment, and 16 questions to measure the wealth index of the households of the participants.

A stadiometer with a sliding headpiece attached was used to measure the height of the adolescent girls before data collection. The height was measured to the nearest 0.1 cm in a standing position with bare feet. After each measurement, the stadiometer was calibrated. And then, the World Health Organization (WHO) Anthroplus 2007 software was used for analyzing anthropometric data in order to identify cases and controls.

Three data collectors and a supervisor were chosen from among people who are knowledgeable about the study area and are capable of doing the task. And then, a one-day intensive and complete training was provided for both data collectors and supervisor. The training was focused on interviewing approaches (techniques) and how to use the EpiCollect5 mobile application to collect data. Then, data was collected by those data collectors via the EpiCollect5 mobile application through a face-to-face interview. During the actual data collection, the supervisor visited each site to follow the data collection, as well as manage any ambiguous activities immediately. The principal investigators were responsible for the coordination of the overall data collection process.

## Data quality control

The data quality was maintained through careful questionnaire design, data collection by well-trained data collectors, and supervision by trained supervisors. The content and face validity of the tool were checked. The questionnaire was pretested on 21 adolescent girls, which was (5%) of the sample of adolescent girls in Woinwuha elementary and secondary schools, to ensure reliability. Feedback from the pre-test was incorporated into the final questionnaire design, and necessary amendments, especially ordering of the questionnaire and removing ambiguous questions were made. Every day, the collected data was reviewed and checked for completeness and consistency by the supervisors as well as the principal investigator.

## Data processing and analysis

Epicollect5 software was used to clean up and code the data before it was downloaded into a CSV spreadsheet. And then, data entry and recoding were done by Epi Data Manager version 4.6 and exported to Statistical Package for Social Science (SPSS) software version 26 for analysis. Descriptive statistics such as frequency, percent, median, and interquartile range were used to describe the study population in relation to relevant variables. A bivariable and multivariable logistic regression analysis was done to identify determinants of stunting among adolescent girls.

First, the association between one independent variable and the dependent variable was evaluated using bivariable logistic regression. Variables with a P- value $\leq 0.25$ in bivariable analysis were candidates for multivariable logistic regression analysis. Factor analysis (Principal component analysis) (PCA) was done to compute wealth index via SPSS version 26. In order to compute the factor analysis, first the variables were changed into a dummy table, and six components were extracted with two eigen values $> 1$. The next two components were extracted, and finally, only one component was extracted and ranked in five quantiles to get the wealth index score.

Multicollinearity was checked using the Variance Inflation Factor (VIF) and Spearman's correlation coefficient test. There was no multicollinearity since all variables had a VIF $< 10$, and the Spearman's correlation coefficient of between -0.7 and +0.7. The Hosmer-Lemeshow goodness-of-fit test was used to test for the model fitness, and a P-value for the Hosmer and Lemeshow test was 0.63. A multivariable logistic regression model was used to identify potential significant determinants of stunting after controlling for all possible potential confounders. And then, variables with a P- value $< 0.05$ at 95% Cl were declared as statistically significant. Finally, the study was presented through statements, figures and tables.

## Ethics approval and consent for participation

The Institutional Review Board (IRB) at Bahir Dar University's College of Medicine and Health Science granted ethical approval under protocol 403/2022. A permission letter was obtained from Bibugn Wereda Education and Health Bureau as well as from each school. Informed assent was obtained from each study participant who was less than 18 years old. A written consent letter was sent to parents/guardians by those participants who had signed on the assent. Written consent was obtained from participants whose age $\geq 18$ years old. Study participants were assured that, there was no physical or emotional harm resulting from participating in the study and were informed about the aim of the study. The participation of the respondents was voluntary, and they were free to withdraw their participation at any time. Participants' confidentiality was strictly held by the research team.

# Result

## Socio demographic factors

A total of 409 adolescent girls (100 cases and 309 controls) participated, with a response rate of 96% for cases and 98.72% for controls. The median age with an interquartile range (IQR) for study participants was 17 (15–18) and 17 (16–18) years for cases and controls, respectively. Majority of cases (75% of them) and controls (50.8%) lived in rural areas. Cases and controls had significant difference in terms of residence ($X^2$ = 17.04, P = 0.001), marital status ($X^2$ = 10.61, P = 0.005), and family size ($X^2$ = 10.22, P = 0.001). On the other hand, cases and controls had no significant differences in terms of age ($X^2$ = 0.29, P = 0.862) (Table 2).

Regarding wealth index of the study participants, cases and controls had a significant difference in terms of the wealth index (X2 = 11.18, P = 0.025). About 29 (29%) of cases and 63

**Table 2. Socio demographic characteristics of adolescent girls in schools of Digo Tsion Town, Northwest Ethiopia, 2022, ($N_T$ = 409, case = 100, control = 309).**

| Variable | Category | Case N (%) | Control N (%) | $X^2$ | P–value |
|---|---|---|---|---|---|
| Age | Early adolescent | 10 (10) | 26 (8.4) | 0.29 | 0.862 |
| | Middle adolescent | 34 (34) | 111 (35.9) | | |
| | Late adolescent | 56 (56) | 172 (55.7) | | |
| Residence | Rural | 75 (75) | 157 (50.8) | 17.04 | 0.001 |
| | Urban | 25 (25) | 152 (49.2) | | |
| Marital status | Single | 88 (88) | 298 (96.7) | 10.61 | 0.005 |
| | Married | 9 (9) | 7 (2.2) | | |
| | Divorced | 3 (3) | 4 (1.1) | | |
| Family size | < 5 members | 19 (19) | 114 (36.9) | 10.22 | 0.001 |
| | ≥ 5 members | 81 (81) | 195 (63.1) | | |
| Educational status of father | Unable to read and write | 48 (48) | 73 (23.6) | 22.49 | 0.001 |
| | Read and write only | 22 (22) | 89 (28.8) | | |
| | Primary school | 11 (11) | 43 (13.9) | | |
| | Secondary school | 8 (8) | 38 (12.3) | | |
| | College/University | 11 (11) | 66 (21.4) | | |
| Occupation of fathers | Farmer | 82 (82) | 185 (59.9) | 23.9 | 0.001 |
| | Daily laborer | 4 (4) | 4 (1.3) | | |
| | Merchant | 2 (2) | 33(10.3) | | |
| | Government employ | 8 (8) | 68 (22) | | |
| | Unemployed | 1 (1) | 7 (2.3) | | |
| | Others** | 3 (3) | 12 (3.9) | | |
| Educational status of mother | Unable to read and write | 54 (54) | 95 (30.7) | 20.90 | 0.001 |
| | Read and write only | 20 (20) | 93 (30.1) | | |
| | Primary school | 11 (11) | 35 (11.3) | | |
| | Secondary school | 7 (7) | 25 (8.1) | | |
| | College/University | 8 (8) | 61 (19.7) | | |
| Occupation of mother | Farmer | 75 (75) | 177 (57.3) | 17.74 | 0.001 |
| | Merchant | 10 (10) | 24 (7.8) | | |
| | Government employ | 4 (4) | 56 (18.1) | | |
| | House wife | 9 (9) | 49 (15.9) | | |
| | Others*** | 2 (2) | 3 (1) | | |

*Grandparents, Ante, Uncle, Siblings

**Driver, Clergy man, Carpenter

***Daily laborer, Waiter

(20.4%) of controls were at the first level (poorest). Thirty-one (31% of cases) and 20.7% of controls were poor, while 10 (10%) of cases and 54 (17.5%) of controls were at the highest level (richest).

## Dietary and nutritional service factors

Nearly half (52%) of the case group used wheat, barely, maize, and potatoes as their usual foods, while 148 (47.9%) of the control group used teff, maize, wheat, and barley. Only two (2%) of cases and four (1.3%) of controls used only teff as their usual food. Regarding regular meal skipping per day, 46 (46% of cases) and 62 (20.1%) of controls skipped their regular meal. The majority of the respondents in both cases, 82 (82%), and controls, 190 (61.5%), used their own products and market-purchased food sources for daily consumption. Approximately 69 (69%) of cases involved drinking coffee or tea immediately while eating a meal, and 63 (63%) consumed it occasionally. More than half of the controls (180, or 54.7%) did not drink at all. Cases and controls had significant differences in terms of meal frequency, usual food type, etc. Using Snack did not show a significant difference (Table 3).

## Personal hygiene, household and environmental-sanitation factors

More than three-quarters (88%) of cases and 304 (98.4%) of controls had latrine facility differences in their house. Thirty-five (35%) of cases and 50 (16.2%) of controls used unimproved water sources for drinking. About two (2%) of the participants from the cases and 11 (3.6%) from the controls used the water purification method. Cases and controls had significant differences in terms of availability of latrines ($X^2$ = 17.92, P = 0.001), drinking water source, and hand washing after using the toilet. Water purification methods and waste disposal sites did not show any significant differences (Table 4).

## Health related and health service factors

Health extension workers visited the homes of approximately 36 (36%) cases and 188 (60.8%) controls. During the home visit, 3 (3%) of cases and 43 (22.9%) of controls received information about adolescent nutrition. More than half (60%) of cases did not receive deworming tablets, while 169 (54.7%) of controls did. The majority of study participants in both cases and controls had seen their first menstruation. Among those, 56 (56%) of cases and 171 (65.5%) of controls saw their first period at a middle adolescent stage. Cases and controls had a significant difference in terms of receiving a deworming tablet ($X^2$ = 5.95, P = 0.015) and a HEW visit ($X^2$ = 18.82, P = 0.001). On the other hand, cases and controls had no significant difference in terms of anemia ($X^2$ = 0.21, P = 0.651), malaria ($X^2$ = 0.912, P = 0.34), chronic illness ($X^2$ = 1.521, P = 0.218), and menstrual status ($X^2$ = 0.321, P = 0.571) (Table 5).

## Determinants of stunting among school adolescent girls

The result of the multivariable analysis indicated that, adolescent girls who ate a low-diversified diet were nearly two (1.99) times more likely to be stunted compared with their counterparts who ate adequately diversified meals (AOR = 1.99, CI [1.06, 3.73]). Adolescent girls living in food insecure households were 2.13 times more likely to be stunted compared with those in food secure households (AOR = 2.13, CI [1.15, 3.93]). Those who drank coffee/tea immediately while eating meals experienced 2.19 times more stunts compared with their counterparts (AOR = 2.19, CI [1.22, 3.95]).

Adolescents who did not get nutritional counseling in school were 2.07 times more likely to be stunted compared with those who got counseling (AOR = 2.07, CI [1.17, 3.66]). Girls with

**Table 3. Dietary and nutritional service information of adolescent girls in schools of Digo Tsion Town, Northwest Ethiopia, 2022, ($N_T$ = 409, case = 100, control = 309).**

| Variables | Category | Case N (%) | Control N (%) | $X^2$ | P-value |
|---|---|---|---|---|---|
| Usual food type in household | Teff only | 2(2) | 4(1.3) | 139.75 | 0.001 |
| | Teff, maize, wheat | 5(5) | 70(22.7) | | |
| | Teff, maize, wheat, barely | 27(27) | 148(47.9) | | |
| | Teff, maize, wheat, barely, potato | 14(14) | 76(24.6) | | |
| | Maize, wheat, barely, potato | 52(52) | 11(3.6) | | |
| Decision maker in household | Mother & father | 24(24) | 76(24.6) | 11.57 | 0.009 |
| | Mother only | 58(58) | 212(68.6) | | |
| | Father only | 8 (8) | 11 (3.6) | | |
| | Others* | 10 (10) | 10 (3.2) | | |
| Source of food for consumption | Market purchase | 17(7) | 115(37.2) | 14.40 | 0.001 |
| | Own product & market purchase | 82(82) | 190(61.5) | | |
| | Own product | 1 (1) | 4 (1.3) | | |
| Meal per day | ≤ Two times | 47 (47) | 64 (20.7) | 25.09 | 0.001 |
| | ≥ Three times | 53(53) | 245(79.3) | | |
| Snack usage | Yes | 44 (44) | 132 (42.7) | 0.01 | 0.913 |
| | No | 56 (56) | 177(57.3) | | |
| Skipping regular meal | Yes | 46 (46) | 62(20.1) | 24.83 | 0.001 |
| | No | 54(54) | 247(79.9) | | |
| Drinking coffee/tea immediately with meal | Not at all | 31(31) | 169(54.7) | 19.32 | 0.001 |
| | Sometimes | 63(63) | 118(38.2) | | |
| | Always | 6(6) | 22(7.1) | | |
| Dietary diversity | ≤ 4 | 66(66) | 108(35) | 28.54 | 0.001 |
| | >4 | 34(34) | 201(65) | | |
| Food security | Secured | 51(51) | 243(78.6) | 27.21 | 0.001 |
| | Insecure | 49(49) | 66(21.4) | | |
| Nutritional counseling | Yes | 43(43) | 208(67.3) | 17.83 | 0.001 |
| | No | 57(57) | 101(32.7) | | |
| Using mass media | Yes | 42(42) | 234(75.7) | 37.64 | 0.001 |
| | No | 58(58) | 75(24.3) | | |

* Grandparents, Antes, Uncle, Siblings, Self, $X^2$ = chi square, Df = degree of freedom

chronic illness were 3.78 times more likely to be stunted than girls who had no chronic illness (AOR = 3.78, CI [1.16, 12.3]). Adolescent girls who were not visited by HEW at their home were nearly two (1.85) times more likely to be stunted compared with their counterparts (AOR = 1.85, CI [1.03, 3.31]) (Table 6).

## Discussion

The aim of this study was to assess the determinants of stunting. As a result, this study found that food insecurity is an independent risk factor for stunting among school adolescent girls, which indicates living in a food insecure household was 2.13 times more likely to be stunted. This finding was consistent with previous studies conducted in Debark [26], Legehida district [27], Afar region and Northeast Ethiopia [33]. The possible explanation might be the fact that food insecurity is the cause of undernutrition in the community. Those individuals living in food insecure households are at risk of stunting and poor health outcomes, which can affect the coming generations [34]. And also, it might be due to food insecurity, individuals are not

**Table 4. Personal hygiene, household and environmental-sanitation information of adolescent girls in schools of Digo Tsion Town, Northwest Ethiopia, 2022, ($N_T$ = 409, case = 100, control = 309).**

| Variables | Category | Case N (%) | Control N (%) | $X^2$ | P-value |
|---|---|---|---|---|---|
| Available latrine facility in house | Yes | 88(88) | 304(98.4) | 17.92 | 0.001 |
| | No | 12(12) | 5(1.6) | | |
| Drinking water source | Tape water | 21(21) | 149(48.2) | 136.7 | 0.001 |
| | Protected dug water | 25(25) | 110(35.6) | | |
| | Protected spring water | 19(19) | - | | |
| | Unprotected dug water | 6(6) | 42(13.6) | | |
| | Unprotected spring water | 26(26) | 7(2.3) | | |
| | Other* | 3(3) | 1(0.3) | | |
| Using water purification method | Yes | 2(2) | 11(3.6) | 0.198 | 0.656 |
| | No | 98(98) | 298(96.4) | | |
| Type of water purification method | Wuha agar | 2(100) | 8(72.7) | 0.709 | 0.701 |
| | Boiling | - | 2(18.2) | | |
| | Others* | | 1(9.1) | | |
| Hand washing before eating | Always | 91(91) | 305(98.7) | 12.18 | 0.001 |
| | Sometimes | 9(9) | 4(1.3) | | |
| Hand washing after using a toilet | Yes | 87(87) | 289(93.5) | 4.34 | 0.037 |
| | No | 13(13) | 20(6.5) | | |
| Using soap for hand washing | Yes | 55(55) | 229(74.1) | 12.11 | 0.001 |
| | No | 45(45) | 80(25.9) | | |
| Separate waste disposal site | Yes | 2(2) | 8(2.6) | 0.11 | 0.740 |
| | No | 98(98) | 301(62.1) | | |

*Surface water, river water

**String through cloth

**Table 5. Health related and health service information of adolescent girls in schools of Digo Tsion Town, Northwest Ethiopia, 2022, ($N_T$ = 409, case = 100, control = 309).**

| Variables | Category | Case N (%) | Control N (%) | $X^2$ | P-value |
|---|---|---|---|---|---|
| Receiving deworm | Yes | 40(40) | 169(54.7) | 5.95 | 0.015 |
| | No | 60(60) | 140(45.3) | | |
| Anemia | Yes | 3(3) | 5(1.6) | 0.21 | 0.651 |
| | No | 97(97) | 304(98.4) | | |
| Malaria | Yes | 2(2) | 15(4.9) | 0.912 | 0.34 |
| | No | 98(98) | 294(95.1) | | |
| Chronic illness | Yes | 8(8) | 13(4.2) | 1.521 | 0.218 |
| | No | 92(92) | 296(95.8) | | |
| First menstruation | Yes | 88(88) | 265(85.8) | 0.321 | 0.571 |
| | No | 12(12) | 44(14.2) | | |
| Age at first menarche | 10–13 | 32(32) | 94(35.5) | 21.45 | 0.001 |
| | 14–16 | 56(56) | 171(65.5) | | |
| Visited by HEW | Yes | 36(36) | 188(60.8) | 18.82 | 0.001 |
| | No | 64(64) | 121(39.2) | | |

HEW = Health Extension Worker

**Table 6. Bivariable and multivariable logistic regression analysis showing determinant of stunting among adolescent girls in schools of Digo Tsion Town, Northwest Ethiopia, 2022, (n = 409).**

| Variable | | Case (%) | Control (%) | COR (95% CI) | AOR (95% CI) |
|---|---|---|---|---|---|
| Residence | Rural | 75(75) | 157(50.8) | 2.9[1.75, 4.81] | 1.32[0.55, 3.16] |
| | Urban | 25(25) | 152(49.2) | 1 | 1 |
| Family size | < 5 members | 19(19) | 195(63.1) | 1 | 1 |
| | ≥ 5 members | 81(81) | 114(36.9) | 2.49[1.44, 4.32] | 1.75[0.89, 3.44] |
| Educational status of fathers | Unable to read and write | 48(48) | 73(23.6) | 3.95[1.89, 8.23] | 1.96[0.66, 5.82] |
| | Read and write only | 22(22) | 89(28.8) | 1.48[0.67, 3.27] | 1.11[0.36, 3.43] |
| | Primary school | 11(11) | 43(13.9) | 1.54[0.61, 3.85] | 1.35[0.39, 4.59] |
| | Secondary school | 8(8) | 38(12.3) | 1.26[0.47, 3.41] | 1.29[0.36, 4.63] |
| | College/University | 11(11) | 66(21.4) | 1 | 1 |
| Educational status of mothers | Unable to read and write | 54(54) | 95(30.7) | 4.33[1.93, 9.74] | 1.01[0.33, 3.11] |
| | Read and write only | 20(20) | 93(30.1) | 1.64[0.68, 3.96] | 0.62[0.19, 2.01] |
| | Primary school | 11(11) | 35(11.3) | 2.4[0.88, 6.52] | 0.99[0.29, 3.36] |
| | Secondary school | 7(7) | 25(8.1) | 2.14[0.69, 6.52] | 1.16[0.30, 4.46] |
| | College/University | 8(8) | 61(19.1) | 1 | 1 |
| Wealth index of households | Poorest | 29(29) | 63(20.4) | 2.49[1.11, 5.56] | 1.04[0.32, 3.34] |
| | Poor | 31(31) | 64(20.7) | 2.62[1.18, 5.82] | 1.29[0.41, 4.11] |
| | Middle | 14(14) | 60(19.4) | 1.26[0.52, 3.07] | 0.48[0.14, 1.64] |
| | Rich | 16(16) | 68(22) | 1.27[0.53, 3.02] | 0.99[0.33, 2.93] |
| | Richest | 10(10) | 54(17.5) | 1 | 1 |
| Meal frequency per day | ≤ Twice times | 47(47) | 64(20.7) | 3.39[2.10, 5.48] | 2.56[0.94, 6.96] |
| | ≥ Three times | 53(53) | 245(79.3) | 1 | 1 |
| Skipping regular meals | Yes | 46(46) | 62(20.1) | 3.39[2.09, 5.49] | 1.01[0.37, 2.75] |
| | No | 54(54) | 247(79.9) | 1 | 1 |
| Drinking coffee/tea with meal | Yes | 69(69) | 140(45.3) | 2.69[1.66, 4.34] | 2.19[1.22, 3.95] |
| | No | 31(31) | 169(54.7) | 1 | 1 |
| Nutritional counsel in school | Yes | 43(43) | 208(67.3) | 1 | 1 |
| | No | 57(57) | 101(32.7) | 2.73[1.72, 4.33] | 2.07[1.17, 3.66] |
| Using mass media | Yes | 42(42) | 243(75.7) | 1 | 1 |
| | No | 58(58) | 75(24.3) | 4.3[2.68, 6.93] | 1.85[0.98, 3.49] |
| Dietary diversity score | ≤ 4 | 66(66) | 108(35) | 3.6[2.25, 5.81] | 1.99[1.06, 3.73] |
| | >4 | 34(34) | 201(65) | 1 | 1 |
| Food security status | Insecure | 49(49) | 66(21.4) | 3.5[2.19, 5.70] | 2.13[1.15, 3.93] |
| | Secure | 51(51) | 243(78.6) | 1 | 1 |
| Drinking water source | Unimproved | 36(36) | 49(15.9) | 2.98[1.79, 4.97] | 1.09[0.54, 2.19] |
| | Improved | 64(64) | 260(84.1) | 1 | 1 |
| Hand washing after toilet | Yes | 87(87) | 289(93.5) | 1 | 1 |
| | No | 13(13) | 20(6.5) | 2.16[1.03, 4.52] | 2.29[0.88, 6] |
| Using soap for hand washing | Yes | 55(55) | 229(74.1) | 1 | 1 |
| | No | 45(45) | 80(25.9) | 2.34[1.46, 3.74] | 1.16[0.63, 2.13] |
| Receiving deworming | Yes | 40(40) | 169(45.3) | 1 | 1 |
| | No | 60(60) | 140(54.7) | 1.81[1.14, 2.86] | 0.74[0.41, 1.35] |
| Chronic illness | Yes | 8(8) | 13(4.2) | 1.98[0.79, 4.93] | 3.78[1.16, 12.3] |
| | No | 92(92) | 296(95.8) | 1 | 1 |
| HEW visit at home | Yes | 36(36) | 188(60.8) | 1 | 1 |
| | No | 64(64) | 121(39.2) | 2.76[1.73, 4.41] | 1.85[1.03, 3.31] |

Note; AOR- Adjusted odds Ratio, COR- Crude Odds Ratio, CI- confidence interval

able to consume sufficient amounts of safe and nutritious food for an active and healthy life, which results in chronic complications of undernutrition [35].

Likewise, eating a low diversified diet (low dietary diversity score) was identified as one of the independent risk factors contributing to stunting. The odds of adolescent girls eating a low diversified diet in the case group were 1.99 times higher than in the counter-control group. This finding is in agreement with studies conducted in Dessie Town [36], Awash Town [7], Gonder and Dembia districts, Northwest Ethiopia [37, 38]. The association of low dietary diversity score and stunting might be due to the fact that low variety foods do not fulfill micronutrient requirements, such as iron, vitamin B12, folate, and other essential requirements for growth resulting in linear growth retardation (stunting) [39]. Furthermore, another possible reason might be that an adequate supply of all essential nutrients has a fundamental importance in satisfying the nutritional requirements for the maintenance of a body's growth, strength, physical work, immunity, and good health.

Drinking coffee/tea immediately while eating a meal is one of the significant determinants of stunting among adolescent girls. The odds of drinking coffee/tea immediately with a meal rather than not drinking were about 2.19 times higher among cases than controls. This finding is inconsistent with a study conducted in Gonder town [40]. The reason for the disagreement might be the difference in the study design, characteristics of the study population, and sample size. Even if this is not clearly reported by previous scholarly articles, the association between drinking coffee/tea immediately while eating a meal and stunting might be due to the caffeine content of coffee and tea slightly reducing calcium and iron absorption, which may inhibit bone growth and linear development in children and adolescents. In addition, taking coffee/tea with meals initiates hiccupping and gastrointestinal disturbance, which in turn decreases adequate meal consumption in adolescents. Drinking coffee/tea also results in sleep disturbance (causes lack of sleep), especially those drank within six hours before sleep time, which results in reduced growth hormone production since it is more abundant during sleep [41]. This indicates that sleeping for a shorter period of time than usual causes growth retardation or stunting.

Did not receive nutritional counseling in school was also identified as a significant independent determinant of stunting among adolescent girls. The odds that the cases did not get nutritional counseling in school was nearly twofold (2.07) higher than those of the controls. This finding is supported by studies conducted in Adwa town, urban Northwest Ethiopia [37, 38] and the low land area of Southern Ethiopia [28]. The possible justification might be that nutritional counseling helps adolescent girls know and understand important information about their healthy eating practice as well as how it could reduce risky behavior (like skipping meals to improve physical posture) [42].

Based on this study's findings, having a known chronic illness was a significant determinant of stunting in adolescent girls. The odds of having a chronic illness were 3.78 times higher in the case group than in the controls group. The possible reason for this association of chronic illness with stunting might be due to the fact that adolescents with chronic illness are at greater risk for eating disorders than adolescents without chronic illness. This further causes chronic malnutrition and developmental delay [43]. According to the WHO report, delayed growth and puberty are common among most adolescents with chronic illnesses. Chronic illness can affect the growth and maturation of adolescents.

This study also revealed that not being visited by health extension workers in the home was independently associated with stunting. The odds of not being visited by a health extension worker in their home among cases were about 1.85 times higher than those of controls. This finding was supported by study conducted in Wolaita and Hadiya zones of southern Ethiopia [28]. The possible reasons for this association of not being visited by health extension workers

and stunting might be the fact that health extension workers provide services on health packages like food hygiene and safety measures, healthy home environment, family nutrition and adolescent reproductive health services. These may be used for the prevention of undernutrition in the community either directly or indirectly [44].

Also, they could presumably provide nutritional counseling that would lead to an increase in nutritional knowledge and a change in behavior to enhance nutrition among family members [45]. Moreover, health extension specialists are essential in advancing adolescent health services in both urban and rural regions. For instance, they collaborate with the Productive Safety Net Program (PSNP), which by feeding students, reduces home food insecurity and expands school feeding programs [46].

## Limitation of the study

This study might have the following limitations: Since the data was collected through an interviewer administered questionnaire, there may be social desirability and recall bias, especially for dietary diversity scores and food security assessments. There might also be misclassification of genetically short-statured adolescent girls as undernourished or stunted (cases).

## Conclusion and recommendation

Stunting among adolescents is influenced by a low dietary diversity score, a food-insecure household, drinking coffee or tea immediately while eating a meal, not receiving nutritional counseling, having a chronic illness, and not being visited by health extension workers at home. Future researchers would do better to undertake prospective studies that are able to include variables such as the amount of coffee or tea consumed per day and institutional factors. Health extension workers are better able to provide nutritional counsel by visiting the community's house. The parents and adolescent better to feed diversified food in the family member.

## Supporting information

**S1 Checklist. The STROBE checklist.**
(DOCX)

**S1 Data. The SPSS raw data for cases in the study.**
(SAV)

## Acknowledgments

First of all, we would like to thank Bahir Dar University for giving us the ethical clearance to carry out this study. Our grateful thanks are also forwarded to the directors of each school for their permission to conduct this study and their cooperation throughout the process. Finally, we would like to thank the data collectors, supervisors, and all study participants for their consent, patience, and informativeness in providing baseline data.

## Author Contributions

**Conceptualization:** Ambaw Abebaw Emrie, Getasew Tesfa, Tamene Fetene Terefe, Bogale Chekole, Fisha Alebel GebreEyesus, Legese Fekede Abza, Selamsew Kindie Nega.

**Data curation:** Ambaw Abebaw Emrie, Yeneneh Ayalew, Fisha Alebel GebreEyesus, Selamsew Kindie Nega.

**Formal analysis:** Ambaw Abebaw Emrie, Getasew Tesfa, Yeneneh Ayalew, Adugnaw Bantie Kebie, Tamene Fetene Terefe, Agerie Aynalem Mewahegn, Bogale Chekole, Legese Fekede Abza, Selamsew Kindie Nega.

**Funding acquisition:** Agerie Aynalem Mewahegn, Legese Fekede Abza.

**Methodology:** Ambaw Abebaw Emrie, Getasew Tesfa, Adugnaw Bantie Kebie, Tamene Fetene Terefe, Agerie Aynalem Mewahegn, Bogale Chekole, Fisha Alebel GebreEyesus, Legese Fekede Abza, Selamsew Kindie Nega.

**Resources:** Yeneneh Ayalew, Adugnaw Bantie Kebie, Agerie Aynalem Mewahegn, Selamsew Kindie Nega.

**Software:** Ambaw Abebaw Emrie, Getasew Tesfa, Yeneneh Ayalew, Adugnaw Bantie Kebie, Tamene Fetene Terefe, Agerie Aynalem Mewahegn, Bogale Chekole, Selamsew Kindie Nega.

**Supervision:** Ambaw Abebaw Emrie, Getasew Tesfa, Yeneneh Ayalew, Adugnaw Bantie Kebie, Tamene Fetene Terefe, Agerie Aynalem Mewahegn, Fisha Alebel GebreEyesus, Legese Fekede Abza.

**Validation:** Tamene Fetene Terefe, Bogale Chekole, Legese Fekede Abza.

**Writing – original draft:** Ambaw Abebaw Emrie, Adugnaw Bantie Kebie, Agerie Aynalem Mewahegn, Bogale Chekole, Fisha Alebel GebreEyesus.

**Writing – review & editing:** Ambaw Abebaw Emrie, Getasew Tesfa, Tamene Fetene Terefe, Selamsew Kindie Nega.

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
