## [Decision Letter · Decision Letter 0]

22 Mar 2023

PONE-D-22-34315Determinants of stunting among adolescent girls in schools of Digo Tsion town, Northwest Ethiopia, 2022: unmatched case control studyPLOS ONE

Dear Dr. Emrie,

Thank you for submitting your manuscript to PLOS ONE. After careful consideration, we feel that it has merit but does not fully meet PLOS ONE’s publication criteria as it currently stands. Therefore, we invite you to submit a revised version of the manuscript that addresses the points raised during the review process.

We look forward to receiving your revised manuscript.

Kind regards,

Dereje Haile, MPH/RH

Academic Editor

PLOS ONE

Journal Requirements:

3. We note that you have referenced (ie. Bibugn Wereda Education office report [22]) which has currently not yet been accepted for publication. Please remove this from your References and amend this to state in the body of your manuscript: (ie “Bibugn Wereda Education office report [Unpublished]”) as detailed online in our guide for authors:

**Additional Editor Comments:**

Reviewer 1

Generally, the concern raised is interested and the whole document needs thorough language revision

1. The recommendation given in the abstract and the main conclusion section are not specific and not raised from the finding of the study. the way the authors state the recommendation is known by default and does not show any expertise effort.

2. introduction of the study is not well organized and needs some revision on content, concept and organization of paragraphs.

. the definition of stunting and adolescent stated under introduction are operational definitions, it is enough to define what stunting is

. it is better to include single issue in a certain paragraph, and it is better and essential to come up with social, economic, and psychological burden of stunting on the community.

3. the title shows the study as it is done in Northwest Ethiopia but under study design and setting section it is located as just north of Ethiopia, make it consistent please.

4. the source and the study population are the same and redundant, and it is possible to include it under single subtitle like 'Source and Study population'.

5. why do not the study exclude participants suffering from other impairments that render their communication ability?

6. what type of interventions was done to protect beneficence of pregnant participants and stunted students, hence they are early to cope up with their pregnancy and to reduce the impact of the stunting, respectively?

7. in order to calculate sample size, the study uses some of recently reported determining factors. I think consistency is the main additional factor to select factors to be used for sample size calculation, I doubt on using appropriate sample size calculation method.

8. why did not the study include variables that may have potential effect on the outcome variable? eg. Grade level of the student

9. better to make name of the group of variables simple and catchy

10. there is nothing stated that for whom the interview was upheld. if the data was collected by interviewing the students, the relevance of the result will get under question mark, since the students at lower class level and the stunted respondents are not mentally competent enough to respond to the questionnaire well.

11. I think that 1 day training for the data collectors is not sufficient enough to make them familiar with the tool and the application used to collect the data.

12. checking completeness of the data in the daily manner was stated both under the data collection tool and procedure, and data quality control section. avoid repetition please.

13. Since the problem has been studied repeatedly especially in Northern part of Ethiopia (Northwest, North, and Northeast) at where the study area of this study found, I think the result is not novel to the reader.

14. the relevance of the study was also compromised by involving too young age and stunted study subjects that are not mentally mature enough to respond to the interview.

15. I think the bar graph incorporated is not that much essential to use and it is enough to describe it with words only.

16. I think it is better to acknowledge the deserved bodies only and make it accurate.

17. please include the specific area at where the interview was proceeded, the stunted students identified, and its status regarding privacy to give them emotional protection.

Reviewer 2

Comments on Determinants of stunting among adolescent girls in schools of Digo Tsion town,

Northwest Ethiopia, 2022: unmatched case control study.

General comment:

It is good to come up with this type of study. However, this manuscript has a lot of issues starting from the language of write up to methodological rigor. The following comments may help to modify the study in the future.

Title: please remove the year

Abstract:

Method and material: Avoid the word “Unmatched” unless it is matched case control no need to mention

Is stadiometer used for anthropometric analysis?

Conclusion:

I would be happy if you could modify your conclusion of the abstract.

Method and material

How did you identify the cases and controls before collecting the data?

What is the difference between study population and source population for this study?

Exclusion criteria

Is it ethical to exclude those girls with spinal curvature, third trimester pregnant, who could not stand properly and walked with a wheelchair were excluded for both cases and controls?

How many of these excluded from the study?

Sampling technique and procedures

How did you identify the cases and controls initially?

Independent variables:

Stunting is a chronic condition but some of your independent variables are a short time event like drinking coffee/tea, visit by HEW’s, menstrual status. How these short time event predict the chronic condition??

How did you measure menstrual status? What did you measure?

Results:

Please remove DF from the tables

Please merge some of the tables

Reviewers' comments:

Reviewer's Responses to Questions

**Comments to the Author**

1. Is the manuscript technically sound, and do the data support the conclusions?

Reviewer #1: No

Reviewer #2: Partly

2. Has the statistical analysis been performed appropriately and rigorously? 

Reviewer #1: Yes

Reviewer #2: Yes

3. Have the authors made all data underlying the findings in their manuscript fully available?

Reviewer #1: Yes

Reviewer #2: No

4. Is the manuscript presented in an intelligible fashion and written in standard English?

Reviewer #1: Yes

Reviewer #2: No

5. Review Comments to the Author

Reviewer #1: Generally, the concern raised is interested and the whole document needs thorough language revision

1. The recommendation given in the abstract and the main conclusion section are not specific and not raised from the finding of the study. the way the authors state the recommendation is known by default and does not show any expertise effort.

2. introduction of the study is not well organized and needs some revision on content, concept and organization of paragraphs.

. the definition of stunting and adolescent stated under introduction are operational definitions, it is enough to define what stunting is

. it is better to include single issue in a certain paragraph, and it is better and essential to come up with social, economic, and psychological burden of stunting on the community.

3. the title shows the study as it is done in Northwest Ethiopia but under study design and setting section it is located as just north of Ethiopia, make it consistent please.

4. the source and the study population are the same and redundant, and it is possible to include it under single subtitle like 'Source and Study population'.

5. why do not the study exclude participants suffering from other impairments that render their communication ability?

6. what type of interventions was done to protect beneficence of pregnant participants and stunted students, hence they are early to cope up with their pregnancy and to reduce the impact of the stunting, respectively?

7. in order to calculate sample size, the study uses some of recently reported determining factors. I think consistency is the main additional factor to select factors to be used for sample size calculation, I doubt on using appropriate sample size calculation method.

8. why did not the study include variables that may have potential effect on the outcome variable? eg. Grade level of the student

9. better to make name of the group of variables simple and catchy

10. there is nothing stated that for whom the interview was upheld. if the data was collected by interviewing the students, the relevance of the result will get under question mark, since the students at lower class level and the stunted respondents are not mentally competent enough to respond to the questionnaire well.

11. I think that 1 day training for the data collectors is not sufficient enough to make them familiar with the tool and the application used to collect the data.

12. checking completeness of the data in the daily manner was stated both under the data collection tool and procedure, and data quality control section. avoid repetition please.

13. Since the problem has been studied repeatedly especially in Northern part of Ethiopia (Northwest, North, and Northeast) at where the study area of this study found, I think the result is not novel to the reader.

14. the relevance of the study was also compromised by involving too young age and stunted study subjects that are not mentally mature enough to respond to the interview.

15. I think the bar graph incorporated is not that much essential to use and it is enough to describe it with words only.

16. I think it is better to acknowledge the deserved bodies only and make it accurate.

17. please include the specific area at where the interview was proceeded, the stunted students identified, and its status regarding privacy to give them emotional protection.

Reviewer #2: Comments on Determinants of stunting among adolescent girls in schools of Digo Tsion town,

Northwest Ethiopia, 2022: unmatched case control study.

General comment:

It is good to come up with this type of study. However, this manuscript has a lot of issues starting from the language of write up to methodological rigor. The following comments may help to modify the study in the future.

Title: please remove the year

Abstract:

Method and material: Avoid the word “Unmatched” unless it is matched case control no need to mention

Is stadiometer used for anthropometric analysis?

Conclusion:

I would be happy if you could modify your conclusion of the abstract.

Method and material

How did you identify the cases and controls before collecting the data?

What is the difference between study population and source population for this study?

Exclusion criteria

Is it ethical to exclude those girls with spinal curvature, third trimester pregnant, who could not stand properly and walked with a wheelchair were excluded for both cases and controls?

How many of these excluded from the study?

Sampling technique and procedures

How did you identify the cases and controls initially?

Independent variables:

Stunting is a chronic condition but some of your independent variables are a short time event like drinking coffee/tea, visit by HEW’s, menstrual status. How these short time event predict the chronic condition??

How did you measure menstrual status? What did you measure?

Results:

Please remove DF from the tables

Please merge some of the tables

6. PLOS authors have the option to publish the peer review history of their article (what does this mean?). If published, this will include your full peer review and any attached files.

Reviewer #1: No

Reviewer #2: No

---

## [Author Response · Author response to Decision Letter 0]

11 Apr 2023

Reviewer 1: We have incorporated all of your suggestion into our revision. They were very helpful. Thank you.

Reviewer 2: We have incorporated all of your suggestion into our revision. Thank you for your help.

---

## [Decision Letter · Decision Letter 1]

19 Sep 2023

PONE-D-22-34315R1Determinants of stunting among adolescent girls in schools of Digo Tsion town, Northwest Ethiopia: unmatched case control studyPLOS ONE

Dear Dr. Emrie,

Thank you for submitting your manuscript to PLOS ONE. After careful consideration, we feel that it has merit but does not fully meet PLOS ONE’s publication criteria as it currently stands. Therefore, we invite you to submit a revised version of the manuscript that addresses the points raised during the review process.

We look forward to receiving your revised manuscript.

Kind regards,

Mulualem Endeshaw

Academic Editor

PLOS ONE

Journal Requirements:

**Comments to the Author**

1. If the authors have adequately addressed your comments raised in a previous round of review and you feel that this manuscript is now acceptable for publication, you may indicate that here to bypass the “Comments to the Author” section, enter your conflict of interest statement in the “Confidential to Editor” section, and submit your "Accept" recommendation.

Reviewer #1: All comments have been addressed

Reviewer #3: (No Response)

2. Is the manuscript technically sound, and do the data support the conclusions?

Reviewer #1: Yes

Reviewer #3: Partly

3. Has the statistical analysis been performed appropriately and rigorously? 

Reviewer #1: Yes

Reviewer #3: No

4. Have the authors made all data underlying the findings in their manuscript fully available?

Reviewer #1: Yes

Reviewer #3: Yes

5. Is the manuscript presented in an intelligible fashion and written in standard English?

Reviewer #1: No

Reviewer #3: No

6. Review Comments to the Author

Reviewer #1: Dear authors,

Thank you for your job, well done.

I think it is difficult to say that all the comments have been addressed and please be careful to use such like absolute words.

1. the justification stated to show the relevance compromization related to interviewing children suffering from stunting is not satisfactory. including information from those study subjects could not improve relevance of the result.

2. were the newly added variables that much helpful and impactful?

Reviewer #3: I want to extend my appreciation to the authors of this manuscript for their consideration of those vulnerable population segments for malnutrition specifically in our country. Here are my general and specific comments and suggestions to the author.

General comments:

1. What special or unique determinants of stunting did you found as compared to different scientific evidences?

2. Your recommendations were not supported by the findings of this study.

Abstract:

1. Show the gaps related to your research problem

2. lines 34-35: The word "face to-face interview" is not clear. did you used qualitative data collection tool? better to rephrase it.

3. the conclusion part of your abstract is copy paste of your result. Better to narrate your result in different ways.

Introduction:

1. You have to clearly indicated any global or national program initiatives to prevent or minimize stunting in adolescents, any success or failure and the contribution of the current study in relation to those program interventions and strategies.

Methods and Materials:

1. lines 126-127: the study unit was adolescent girl not girls whether she was grouped under control or case

2. lines 128-132: revise your inclusion criteria. the one you stated was already included by default.

3. Lines 134-135: Also revise your exclusion criteria. why you exclude girls "spinal curvature?, who used wheelchair?" There is alternative means of height measurement for them. Did you included 1st and 2nd trimester pregnant?? why you excluded third trimester?

4. Lines 197-199: Make "Wealth index" measurable. the one you stated as the operational definition of wealth index is not appropriate. After you derived the scores, How did you measured it?

5. Data collection tool and procedure is too shallow. Explain the data collection tool and detail procedure for each of your independent variables like; anemia, dietary diversity, food security, presence or absence of malaria, chronic illness, and others.

6. How did you directly compute VIF in logistic regression model??

7. You have to describe in detail the procedure while you run Principal component analysis (PCA) like: what assumptions did you checked? what rotation did you used? How many components were extracted? what was the cut value for eigenvalues? How did you inter-prated the results? Finally how did you measured wealth index based on the extracted components??

Results:

1. Table 3: the dietary diversity was not presented as stated on the operational definition.

2. There was no any findings related to the wealth status of the study respondents. why??

3. Paragraph on lines 329-334 was not needed since it was stated on the next paragraph. You have to begin the next paragraph which was on line 335, as: "The result of the multivariable analysis indicated that, Adolescent girls who ate a low-diversified diet were nearly two (1.99) times more likely to be stunted compared with their counterparts who ate adequately diversified meals (AOR = 1.99, CI [1.06, 3.73]). --------------------------------".

4. under the limitations of your study statement on lines 419-421 needs revision. "there is no anthropometric measurement scale for such a group of individuals." Are you sure??????? Think over it.

5. Under recommendation, why you didn't consider institutional factors?? If you are stated as there is no any anthropometric measurement scale for individuals with spinal curvature, how did you recommended future researcher to undertake studies that are able to include those individuals with spinal curvature???

---

## [Author Response · Author response to Decision Letter 1]

23 Sep 2023

For reviewer I: We have incorporated all of your suggestions into our revision. They were very helpful. Thank you!

For reviewer III: We have incorporated all of your suggestions into our revision. They were very helpful. Thank you for your help!

---

## [Decision Letter · Decision Letter 2]

25 Jan 2024

PONE-D-22-34315R2Determinants of stunting among adolescent girls in schools of Digo Tsion town, Northwest Ethiopia: unmatched case control studyPLOS ONE

Dear Dr. Emire,

Thank you for submitting your manuscript to PLOS ONE. After careful consideration, we feel that it has merit but does not fully meet PLOS ONE’s publication criteria as it currently stands. Therefore, we invite you to submit a revised version of the manuscript that addresses the points raised during the review process.

I share almost all comments raised  by  all the reviewersSpecifically I do have the followings: Some determinants are vague ( not getting nutrition  counsel), which chronic illness was considered in this study? Data need to be considered as plural word throughout the manuscriptTh citation for dietary diversity measurement is not there . Nothing mentioned about DD in ref. #22; Authors need to use the standard citation. How many food groups were considered in this study? How data quality related to height measurement was ascertained? Did the authors do standardization ? This needs to be discussed as a limitation as well. Which of the percentages were used in result presentation? Row or column?It is unfair to include 18 variables at the same time in the multivariable model which could enhance the probability of multicollinearity and model instability Do not report p-value alongside AOR (Table 6) Key findings need to be paraphrased in the first paragraph of the discussion before commencing the discussion of each.Implication of each key finding needs to be discussed in addition to theoretical discussionAll grey literature need to traceable using URL and date accessed.For instance ref. #1&4 are wrongly cited .The author's name is incorrect and also for most of them In sum, the manuscript need to be further refined to the desired level of the journal and scientific rigor.

We look forward to receiving your revised manuscript.

Kind regards,

Gudina Egata, PhD in Public Health

Academic Editor

PLOS ONE

Reviewers' comments:

Reviewer's Responses to Questions

**Comments to the Author**

1. If the authors have adequately addressed your comments raised in a previous round of review and you feel that this manuscript is now acceptable for publication, you may indicate that here to bypass the “Comments to the Author” section, enter your conflict of interest statement in the “Confidential to Editor” section, and submit your "Accept" recommendation.

Reviewer #3: (No Response)

Reviewer #4: (No Response)

2. Is the manuscript technically sound, and do the data support the conclusions?

Reviewer #3: No

Reviewer #4: Yes

3. Has the statistical analysis been performed appropriately and rigorously? 

Reviewer #3: (No Response)

Reviewer #4: Yes

4. Have the authors made all data underlying the findings in their manuscript fully available?

Reviewer #3: (No Response)

Reviewer #4: No

5. Is the manuscript presented in an intelligible fashion and written in standard English?

Reviewer #3: (No Response)

Reviewer #4: Yes

6. Review Comments to the Author

Reviewer #3: Almost all of the comments raised during the first round of review are not fully addressed and still with major methodological and result interpretation issues.

General comments

1. What special or unique determinants of stunting did you found as compared to different

scientific evidences? But the variables indicated by the author were not unique findings

2. Your recommendations were not supported by the findings of this study.

Abstract:

1. On lines 34-35: Ways of administration of structured questionnaires to the respondents was not clear.

Introduction:

1. You have to clearly indicate any global or national program initiatives to prevent or minimize stunting in adolescents, any success or failure and the contribution of the current study in relation to those program interventions and strategies. The global or national program initiatives stated on lines 99-102 is too shallow. Better to explain it in relation to your topic of study with respect to their any success or failure and the contribution of the current study in relation to those program interventions and strategies. Better to use evidences from Global Nutrition Targets and the current Ethiopian Food and Nutrition Policy directions. This issue is still not correctly addressed.

Methods and Materials:

Response to Reviewers

1. On lines 126-127: the study unit was adolescent girl not girls whether she was grouped under control or case and this issue is still not correctly addressed

2. On lines 128-132: revise your inclusion criteria. The one you stated was already included by default. This issue is still not correctly addressed

3. On Lines 134-135: Also revise your exclusion criteria. Why you exclude girls "spinal

curvature? Who used wheelchair?" There is alternative means of height measurement for

them. Did you include 1st and 2nd trimester pregnant?? Why you excluded third trimester? The response provided by the author is not supported by any scientific evidences and still this issue is not correctly addressed

4. On lines 197-199: Make "Wealth index" measurable. The one you stated as the operational

Definition of wealth index is not appropriate. After you derived the scores, how did you

Measured it? Operational definition is not same as dictionary definition of terms and the issue is still not addressed. Thus this issue is still not correctly addressed.

5. Data collection tool and procedure is too shallow. Explain the data collection tool and

detail procedure for each of your independent variables like; anemia, dietary diversity,

food security, presence or absence of malaria, chronic illness, and others. But those issues are still not correctly addressed

6. How did you directly compute VIF in logistic regression model?? Still this issue is not correctly addressed

7. You have to describe in detail the procedure while you run Principal component analysis

(PCA) like: what assumptions did you checked? what rotation did you used? How many

components were extracted? what was the cut value for eigenvalues? How did you inter-

prated the results? Finally, how did you measured wealth index based on the extracted

components?? Still this issue is not correctly addressed

Results:

1. Table 3: the dietary diversity was not presented as stated on the operational definition.

Answer: I have already corrected. But still this issue is not correctly addressed.

4. Under the limitations of your study statement on lines 419-421 needs revision. "There is no anthropometric measurement scale for such a group of individuals." Are you sure? But still this issue is not correctly addressed.

5. under recommendation, why you didn't consider institutional factors?? If you are stated as there is no any anthropometric measurement scale for individuals with spinal curvature, how did you recommended future researcher to undertake studies that are able to include those individuals with spinal curvature? But still this issue is not correctly addressed

Reviewer #4: Dear authors Your title is very interesting and focused on vulnerable age groups. You have chosen a topic that is relevant and important.

The revised version of the paper is better than the previous one, but I still noticed some parts of the paper that need modification. Please check the following suggestions and make the necessary changes.

You have not addressed the questions raised by the third reviewer in the abstract section of your paper. The reviewer asked you to clarify the following:

“1. Show the gaps related to your research problem

2. lines 34-35: The word "face to-face interview" is not clear. did you used qualitative data

collection tool? better to rephrase it.

3. the conclusion part of your abstract is copy paste of your result. Better to narrate your result indifferent ways.” You should have respond it appropriately.

“5. Data collection tool and procedure is too shallow. Explain the data collection tool and detail procedure for each of your independent variables like; anemia, dietary diversity, food security, presence or absence of malaria, chronic illness, and others.” You have no corrections for this question, but it is very important and affects the quality of your paper.

Reviewer #3 raised the question, ‘6. How did you directly compute VIF in the logistic regression model?’ but you have not addressed it well. As you know, the standard error is the best way to see the collinearity of categorical data, so my suggestion is to correct it accordingly.

Pleas correct your case definition it is not correct. Please check your reference again. The correct one is

“Case: is defined based on the world health organization (WHO) 2006 reference population. Children whose height for age z score for height ≤ − 2 SD while.

Controls: is children whose z score for height > -2 SD to z score for height +2 SD.”

Recent evidence shows that there are causes of stunting in addition to nutrition-related factors, such as psychological stimulation, which this study did not include. So, how do you address this?”

Please refer B. Bogin, Patterns of Human Growth Available from: https://www.cambridge.org/core/product/identifier/9781108379977/type/book.

In your ‘Conclusion and Recommendation’ section, you recommend that ‘Future researchers would do better to undertake prospective studies on the amount of coffee or tea consumed per day and institutional factors.’ However, your findings do not pertain to the amount of coffee or tea consumed per day. They focus on the timing of consumption, meaning whether it is immediately after a meal or not. So, why do you recommend about the amount of coffee or tea consumed per day? Please correct this and revise your conclusion section based on your findings.

7. PLOS authors have the option to publish the peer review history of their article (what does this mean?). If published, this will include your full peer review and any attached files.

Reviewer #3: No

Reviewer #4: No

---

## [Author Response · Author response to Decision Letter 2]

19 Feb 2024

Thanks for your constructive comments! We had incorporated all of your comments, and we had amended our manuscript accordingly. 

Thank you again!!

---

## [Decision Letter · Decision Letter 3]

29 Jul 2024

PONE-D-22-34315R3Determinants of stunting among adolescent girls in schools of Digo Tsion town, Northwest Ethiopia: unmatched case control studyPLOS ONE

Dear Dr. Emrie,

Thank you for submitting your manuscript to PLOS ONE. After careful consideration, we feel that it has merit but does not fully meet PLOS ONE’s publication criteria as it currently stands. Therefore, we invite you to submit a revised version of the manuscript that addresses the points raised during the review process.

We look forward to receiving your revised manuscript.

Kind regards,

Abu Sayeed, MSc

Academic Editor

PLOS ONE

Reviewers' comments:

Reviewer's Responses to Questions

**Comments to the Author**

1. If the authors have adequately addressed your comments raised in a previous round of review and you feel that this manuscript is now acceptable for publication, you may indicate that here to bypass the “Comments to the Author” section, enter your conflict of interest statement in the “Confidential to Editor” section, and submit your "Accept" recommendation.

Reviewer #3: (No Response)

Reviewer #5: All comments have been addressed

2. Is the manuscript technically sound, and do the data support the conclusions?

Reviewer #3: No

Reviewer #5: Partly

3. Has the statistical analysis been performed appropriately and rigorously? 

Reviewer #3: No

Reviewer #5: Yes

4. Have the authors made all data underlying the findings in their manuscript fully available?

Reviewer #3: (No Response)

Reviewer #5: Yes

5. Is the manuscript presented in an intelligible fashion and written in standard English?

Reviewer #3: No

Reviewer #5: Yes

6. Review Comments to the Author

Reviewer #3: Abstract

1. Line 36: Interview is the data collection tool used for qualitative study approach. So how did you use it for your data collection?

2. Lines 49-53: better to show the implications of your findings under the conclusion parts of your abstract.

Methods:

1. Exclusion criteria:

Why you exclude girls "spinal curvature? Who used wheelchair?" There is alternative means of height measurement for them. Did you include 1st and 2nd trimester pregnant?? Why you excluded third trimester? Still the response given for this issue is not scientific.

You have to use alternative height measurement for those individuals who had accidental injury to their lower back and walk with wheelchair. In addition it is also possible to identify women with suspected pregnancy without using ultrasound in the second trimester. Thus your justification is not scientific and not logical.

2. Lines 2003-206: Make "Wealth index" measurable. The one you stated as the operational definition of wealth index is not appropriate and not measurable. After you derived the scores, how did you measure it?

3. Lines 208-229: Data collection tool and procedure is too shallow. Explain the data collection tool and detail procedure for each of your independent variables like; anemia, dietary diversity, food security, presence or absence of malaria, chronic illness, and others. Still those issues are not fully addressed on your manuscript.

4. Lines 248-252: You have to describe in detail the procedure while you run Principal Component analysis (PCA) like: what assumptions did you checked? What type of rotation did you used? How many components were extracted? What was the cut value for eigenvalues? How did you interpret the results? Finally, how did you measure wealth index based on the extracted components?? Still all those issues are not fully amended which are very crucial.

Reviewer #5: The revised manuscript is interesting and author incorporated most of the suggestions as raised by the reviewers specifically reviewer#3 but it is hard to read the modification in the last revised text. However, some of the points need to be addressed in relation to this research work as-

1. Is it case-control study or a cross-sectional study?

2. More detailing will be required in the methodology portion.

3. Why stunting is important index for assessing undernutrition and public health problems among adolescent children instead of BMI for age?

4. Have the authors gone through the article - Hermanussen, M., & Scheffler, C. (2024). Stop stunting—A misguided

campaign by well-meaning nutritionists. American Journal of Human Biology, e24068. https://doi.org/ 10.1002/ajhb.24068

7. PLOS authors have the option to publish the peer review history of their article (what does this mean?). If published, this will include your full peer review and any attached files.

Reviewer #3: No

Reviewer #5: No

---

## [Author Response · Author response to Decision Letter 3]

1 Aug 2024

Thank you for your suggestions and questions. We have corrected the comments and answered your questions accordingly. Thank you again!

---

## [Editor Report · Decision Letter 4]

13 Aug 2024

Determinants of stunting among adolescent girls in schools of Digo Tsion town, Northwest Ethiopia: unmatched case control study

PONE-D-22-34315R4

Dear Dr. Ambaw Abebaw Emrie,

We’re pleased to inform you that your manuscript has been judged scientifically suitable for publication and will be formally accepted for publication once it meets all outstanding technical requirements.

Kind regards,

Abu Sayeed, MSc

Academic Editor

PLOS ONE
---

## [Editor Report · Acceptance letter]

30 Aug 2024

PONE-D-22-34315R4 

PLOS ONE

Dear Dr. Emrie, 

I'm pleased to inform you that your manuscript has been deemed suitable for publication in PLOS ONE. Congratulations! Your manuscript is now being handed over to our production team.

Kind regards, 

on behalf of

Mr. Abu Sayeed 

Academic Editor

PLOS ONE